# Topographical Anatomy of the Rhesus Monkey (*Macaca mulatta*)—Part II: Pelvic Limb

**DOI:** 10.3390/vetsci10030172

**Published:** 2023-02-21

**Authors:** Christophe Casteleyn, Nina Robin, Jaco Bakker

**Affiliations:** 1Department of Morphology, Medical Imaging, Orthopedics, Physiotherapy and Nutrition, Faculty of Veterinary Medicine, Ghent University, Salisburylaan 133, 9820 Merelbeke, Belgium; 2Comparative Perinatal Development, Department of Veterinary Medicine, Faculty of Pharmaceutical, Biomedical and Veterinary Medicine, University of Antwerp, Universiteitsplein 1, 2610 Wilrijk, Belgium; 3Animal Science Department, Biomedical Primate Research Centre, Lange Kleiweg, 161, 2288 GJ Rijswijk, The Netherlands

**Keywords:** anatomy, topographical anatomy, rhesus monkey, pelvic limb

## Abstract

**Simple Summary:**

The rhesus monkey (*Macaca mulatta*) is one of the most investigated nonhuman primate species in biomedical research since its anatomy and physiology resemble those of humans. This manuscript fulfills the researcher’s and veterinarian’s need for detailed anatomical data on the rhesus monkey’s pelvic limb. Several rhesus monkey cadavers were dissected to study the anatomy of the muscular, circulatory and peripheral nerve systems of the pelvic limb in relation to each other. The anatomical structures are textually described and illustrated by means of numerous detailed colored images.

**Abstract:**

The rhesus monkey (*Macaca mulatta*) is a widely used model in biomedical research because its anatomy and physiology bear many similarities to those of humans. Extensive knowledge of the anatomy of this nonhuman primate species is not only required for the correct interpretation of obtained research data but also valuable for the welfare of captive individuals housed in, e.g., zoos. As anatomical publications on the rhesus monkey are hardly available, outdated and provide only line drawings or black-and-white photographs, the anatomy of the rhesus monkey was readdressed in this study. The various anatomical structures are described in relation to each other topographically per hindlimb region. The hip region, the upper limb, the knee, the lower limb and the foot are described from various perspectives. The structures that are visible in the different layers, from the superficial to the deepest layer, were photographed. Although the anatomy of the hindlimbs of rhesus monkeys and humans are remarkably similar, various subtle dissimilarities have been observed. Consequently, an open-access publication that focuses on the anatomy of the rhesus monkey would be highly valued by both biomedical researchers and veterinarians.

## 1. Introduction

Wild populations of rhesus monkeys (*Macaca mulatta*) live in the southeastern parts of Asia, but this species can also be found in research facilities worldwide, as it is one of the most studied nonhuman primates [1,2]. They serve as animal models in toxicity studies, aid in unraveling the pathogenesis of various diseases and are key in the development of many vaccines against, e.g., Malaria, West Nile virus and H5N1 Influenza [3,4]. The genome of the rhesus monkey is 93.5% similar to that of humans [5]. Consequently, the two species have many phenotypical similarities [6].

Rhesus monkeys that are housed in zoos and primate centers often receive drugs, including anesthetics, by means of remote methodologies, such as darting. However, this intervention can cause trauma, ranging from minor lacerations to fractured bones. Another restraint technique relies on the use of a “restraint” or “squeeze” cage, in which the back is gently pulled forward to allow injections. This not only induces high stress levels but also can provoke mechanical muscle damage when the animal is not cooperative.

To ensure the safety of intramuscular (IM), subcutaneous (SC) and intravenous (IV) injections and related practices, anatomical knowledge is mandatory. In addition, it is essential to have a good knowledge of the anatomy of the rhesus monkey to diagnose and treat observed locomotor problems related to the lower limbs (membrum pelvinum, pelvic limb, hindlimb or hindleg). Unfortunately, the literature on the anatomy of the rhesus monkey is scarce. As the book by Hartman and Straus [7] contains only line drawings, it fails to represent the authentic condition. The atlas by Berringer et al. [8] contains only black-and-white pictures that lack clarity. In the recent book chapter by Casteleyn and Bakker [9], the anatomy of the rhesus monkey is illustrated with many color images. Nevertheless, it may be more valuable to anatomists than to veterinary practitioners since the anatomy is described per system and not topographically. This could limit the use of the book chapter during surgical interventions, e.g., when wounds have to be tended to [10].

The objectives of this study were to describe the anatomy of the hindlimb of the rhesus monkey in a topographical manner and to identify the similarities and differences between the anatomy of the human leg and that of the rhesus monkey hindlimb. Therefore, the various anatomical systems are described, per region, in relation to each other. The textual descriptions are illustrated with numerous full-color images, thus making this article an extremely useful reference work for researchers and veterinarians.

## 2. Materials and Methods

### 2.1. Animals

A total of five adult rhesus monkey (*Macaca mulatta*) cadavers, three males and two females with an age range of 5–12 years, were used in this study. The animals had been used in studies at the Biomedical Primate Research Centre (BPRC), Rijswijk, The Netherlands. Only animals that could not be properly treated or were experiencing an unacceptable level of pain, stress or distress (specified humane endpoints) and therefore had to be euthanized were used for this study. Animals were fasted overnight and were sedated by IM injection of 10 mg/kg ketamine hydrochloride (ketamine 10%^®^, Alfasan Nederland B.V., Woerden, The Netherlands) combined with 0.05 mg/kg medetomidine hydrochloride (Sedastart^®^, AST Farma B.V., Oudewater, The Netherlands). Subsequently, 70 mg/kg pentobarbital (Euthasol^®^ 20%; AST Farma B.V.) was injected IV (v. saphena parva). The cadavers were frozen at −18 °C and transported to the Laboratory of Morphology at Ghent University, Belgium, where they were thawed prior to the anatomical investigation.

### 2.2. Dissection and Imaging

Both pelvic limbs of each rhesus monkey were subjected to anatomical dissection. As such, the presented results are based on ten pelvic limbs. However, photographs were only taken of the left pelvic limbs, as this is custom in most anatomical books and atlases. To this purpose, a Canon EOS 450D body (Canon Inc., Tokyo, Japan) combined with a Canon EF-S 18–200 mm f/3.5–5.6 IS lens (Canon Inc.) was employed. The editing of photographs was performed using GIMP 2.10.30 (www.gimp.org accessed on 3 January 2023) and included cropping, adjusting the lighting, optimizing the color temperature and providing a plain black background.

In order to facilitate the dissection of the blood vessels, latex injection was performed. The arterial system of the pelvic limb was filled with red-colored latex (V-sure Eco Latex, Vosschemie Benelux, Belgium) by orthograde injection into the femoral artery. The superficial venous system was filled with blue-colored latex. To this purpose, the saphenous veins (v. saphena magna and v. saphena parva) were injected in the orthograde direction at the level of the tarsus. A guide to, among others, latex injection to visualize hollow structures in veterinary anatomy is available in the literature [11]. In contrast to the recommendation that fresh specimens be preferred for casting, defrosted cadavers were used in the present study. It was unfortunately practically unfeasible to cast fresh cadavers. However, since the goal was to visualize the larger blood vessels and not the smaller tributaries, the here-described approach was regarded as adequate.

### 2.3. Anatomical Terminology

The terminology that is used throughout this article is based on the *Nomina Anatomica Veterinaria* [12]. Since this nomenclature focuses on domestic animals, it does not provide specific terms for the rhesus monkey.

In the aforementioned publications on the anatomy of rhesus monkeys [11,12], human terminology was used. Unfortunately, not all human terms are applicable to or appropriate for the rhesus monkey since the two species are not anatomically identical. This is confusing and prevents the comparison between different anatomical functionalities. For clarity, we opted to use veterinary nomenclature and have put any potential alternative or human terms between square brackets. The latter terms were derived from the anatomy publications of Barone [13,14,15], in which, besides domestic mammals, the human is also described, or from a human anatomy atlas [16].

Latin terms are used in the figure legends as well as the first time a structure is mentioned in the text. However, to increase readability, English terminology is used to further detail the structures.

## 3. Results

In this section, the results of the anatomical dissections are presented per region (Table 1). First, the hip region (regio pelvis), where the pelvic limb relates to the pelvis by means of the hip joint and the junctional musculature, is discussed. Then, the upper hindlimb (regio femoris), the knee (regio genus and regio poplitea), the lower hindlimb (regio cruris) and finally the ankle (tarsus) and foot (pes) follow. The blood vessels, i.e., arteries and veins, and nerves that are in relation to the surrounding muscles are reviewed simultaneously. Per region, various views, such as a lateral, medial, ventral or dorsal view, can be seen. Specifically for the foot, dorsal and plantar views are presented. In the present work, the origins and insertions of the described muscles are given, but their functions are not mentioned since these were recently described [9]. No variations were observed between the five cadavers regarding the anatomical structures. However, the hindlimbs of the male animals were more heavily built since their cadavers were heavier than those of the female animals.

### 3.1. Hip Region

#### 3.1.1. Lateral Approach

After skinning the lateral side of the hip region, the large m. gluteus superficialis [m. gluteus maximus] is obvious on the superficial layer (Figure 1A). This muscle arises from the lumbar fascia and the first three caudal vertebrae. Its tendon is inserted into the fascia lata and the greater trochanter of the femur. The m. tensor fasciae latae, which keeps the fascia lata under tension and, as such, acts as a knee extensor, is located cranial to the superficial gluteus muscle. The origins of this muscle are the ilium and the fascia overlying the m. gluteus medius (fascia glutea). The latter muscle is already partly visible in between the superficial gluteus muscle and the tensor fasciae latae muscle.

Since the m. gluteus medius is located, for the most part, deep to the m. gluteus superficialis and the m. tensor fasciae latae, it can be better distinguished after these muscles have been retracted (Figure 1B). It can now be seen that the gluteus medius muscle arises from the lateral surface of the wing of the ilium. Its insertion is into the greater trochanter of the femur. The m. iliopsoas, which inserts into the lesser trochanter of the femur is located deep to the m. tensor fasciae latae in the proximity of the origin of the m. rectus femoris. Caudal to the gluteus medius muscle arise the n. ischiadicus, the n. gluteus caudalis superficialis [n. gluteus inferior superficialis] and the n. cutaneus femoris caudalis [n. cutaneus femoris posterior] that run deep to this muscle. By detaching the m. vastus lateralis from the m. biceps femoris, the n. ischiadicus can be followed in a distal direction onto the m. obturatorius internus and the m. quadratus femoris. The m. vastus lateralis originates from the greater trochanter of the femur and inserts into the base of the patella. The m. biceps femoris arises from the ischial tuberosity, which, in the rhesus monkey, is covered by the keratinized callositas ischii. Its thin aponeurosis is inserted into the fascia cruris. Intramuscular injections can be administered in this muscle. The m. obturatorius internus has its origin on the obturator membrane and the bone surrounding the obturator foramen on its dorsal side. The tendon is inserted into the intertrochanteric fossa. The m. quadratus femoris runs from the sciatic tuberosity to the lesser femoral trochanter. The n. gluteus caudalis runs toward the m. gluteus superficialis to innervate this muscle. The n. cutaneus femoris caudalis courses caudally toward the subcutis onto the tendon of the m. obturatorius internus.

The m. piriformis and the m. gluteus profundus [m. gluteus minimus] become visible after the transection of the m. gluteus medius (Figure 1C). The m. piriformis, which is located immediately caudal to the m. gluteus profundus, originates from the sacroiliac joint and the first caudal vertebra, while it inserts into the greater trochanter of the femur. The m. gluteus profundus has its origin on the dorsal aspect of the ilium and inserts into the greater femoral trochanter as well. The n. gluteus cranialis [n. gluteus superior] runs deep to the m. gluteus medius and superficial to the m. gluteus profundus and thus in between the two.

The origins of the above-mentioned nerves, called the lumbosacral plexus, can be identified after the m. piriformis is transected and the proximal and distal muscle stumps are retracted (Figure 1D). In this layer, the a. pudenda interna and the n. pudendus, which both run toward the perineum, can be recognized deep to the m. obturatorius internus.

Subsequently, the m. gluteus profundus was transected and its muscle stumps were removed. In addition, the n. ischiadicus was laterally retracted (and fixated over the greater trochanter of the femur), allowing for the study of the remaining deep structures. The tendon of the m. obturatorius internus was lifted by means of tweezers (Figure 1E). After transection, the mm. gemelli became visible (Figure 1F). They originate from the ischium, and their tendons join the tendon of the m. obturatorius internus to insert into the greater trochanter of the femur. Finally, the corpus ossis ilii remains.

#### 3.1.2. Medial Approach

The most superficial muscles that become visible after skinning the medial side of the hip region are the m. sartorius and the m. gracilis (Figure 2A). Together, they delineate Scarpa’s triangle (trigonum femorale). The m. gracilis is a broad muscle that starts from the pelvic symphysis and attaches to the craniomedial aspect of the proximal third of the tibia. The m. sartorius is a long, slender muscle that arises from the cranioventral spine of the ilium. It inserts into the medial side of the proximal third of the tibia. When this muscle is slightly retracted in the cranial direction, a portion of the m. vastus medialis, the m. iliopsoas, the m. adductor longus and the m. semimembranosus can be identified from cranial to caudal. The m. vastus medialis arises from the lesser trochanter of the femur and inserts into the base of the patella. The origin of the m. adductor longus is the pelvic symphysis. It is located lateral (deep) to the gracilis muscle and inserts medially, halfway along the femur. The semimembranosus muscle consists of the smaller and more lateral semimembranosus proprius muscle and the larger and more medial semimembranosus accessorius muscle. Both originate caudally on the sciatic tuberosity. The semimembranosus proprius muscle is inserted medially on the tibial tuberosity, while the accessory semimembranosus muscle is broadly inserted more proximally, at the level of the medial femoral condyle. These muscles can be well distinguished in Figure 2B. The a. femoralis runs within the femoral trigonum together with the v. femoralis. The initial trajectory of the n. femoralis is cranial to the homonymous artery and vein and superficial (medial) to the m. iliopsoas. It joins the femoral artery and vein a few centimeters distal to the base of the femoral trigonum. The ln. inguinalis superficialis is located on the cranial side of the femoral artery and vein. Proximal and distal to this lymph node, the ramified a. circumflexa femoris lateralis branches off the a. profunda femoris. Distal to this artery, a muscular branch of the femoral artery that supplies the m. quadriceps, i.c., the vastus medialis, can be observed.

The m. sartorius and the m. gracilis are subsequently transected near their insertions and retracted (Figure 2B). From cranial to caudal, the following muscles can now be identified: the m. vastus medialis, the m. adductor longus, the m. adductor magnus (partially), both parts of the m. semimembranosus, i.e., the m. semimembranosus accessorius and the m. semimembranosus proprius, and finally the m. semitendinosus. The m. adductor magnus is composed of two parts that individually attach to the proximocaudal part of the femoral diaphysis. Their origins are the pelvic symphysis and the sciatic tuberosity, respectively. The m. semitendinosus arises from the sciatic tuberosity, just caudal to the origin of the biceps femoris muscle. The tendon is located superficial to the semimembranosus muscle and attaches to the medial surface of the tibial shaft, deep to the tendon of the gracilis muscle. On a medial view and after the m. gracilis has been retracted, the muscular belly of the m. biceps femoris bulges caudal to the semitendinosus muscle. This muscle is, however, more visible via a lateral approach. The m. pectineus can be seen when the m. adductor longus is transected. This short, fusiform muscle runs from the pecten pubis to the medioproximal aspect of the femur.

Retracting both parts of the m. semimembranosus together with the m. semitendinosus allows for visualizing the two muscle bellies of the m. adductor magnus (Figure 2C). In between this muscle and the m. biceps femoris courses the a. circumflexa femoris medialis toward the hamstring muscles (m. biceps femoris, m. semitendinosus and m. semimembranosus). The a. profunda femoris forms the transition from the a. iliaca externa to the a. femoralis. It runs deep toward the caudal aspect of the upper leg. The n. ischiadicus can be observed on the caudal side of the caudal muscle belly of the adductor magnus muscle.

Retracting the stumps of the m. adductor longus, as well as transecting and retracting both muscle bellies of the m. adductor magnus, allows for identifying the m. pectineus, the m. adductor brevis and the m. quadratus femoris from cranial to caudal (Figure 2D). The short adductor muscle, which, like the adductor magnus muscle, consists of two bellies, starts just ventral to the foramen obturatum and attaches to the medioproximal aspect of the femur. The quadratus femoris muscle runs from the sciatic tuberosity to the lesser femoral trochanter. The m. obturatorius externus is now partly visible deep to the adductor brevis and quadratus femoris muscles. The sciatic nerve comes in sight caudodistal to the quadratus femoris muscle. A few centimeters distal to this muscle, its division into the cranial n. fibularis communis (formerly known as n. peroneus communis) and the caudal n. tibialis is apparent. The a. circumflexa femoris medialis runs medial to the insertion of the m. adductor brevis into the femoral shaft toward the m. biceps femoris.

When the m. pectineus and the m. adductor brevis are transected and their stumps are retracted, the entire m. obturatorius externus can be distinguished (Figure 2E). This muscle arises from the obturator membrane and the bone surrounding the obturator foramen on its dorsal side. The tendon is inserted into the intertrochanteric fossa. A branch from the n. obturatorius emerges in between this muscle and the m. quadratus femoris to innervate the m. pectineus and the m. adductor brevis.

In Figure 2F,G, all stumps of the previously transected and retracted muscles are removed. The insertion of the m. iliopsoas and the m. quadratus femoris into the lesser femoral trochanter and the insertion of the m. obturatorius externus into the intertrochanteric fossa are noticeable. When the origin of the latter muscle is transected and the muscle is retracted, the edge of the obturator foramen becomes visible. A branch of the n. obturatorius can be seen penetrating this muscle. The stem of this nerve initially runs parallel to the m. iliopsoas in the direction of the obturator foramen, through which it reaches the adductor musculature of the hindlimb. At last, it should be mentioned that the m. quadriceps, which is elaborated below, can be observed on the cranial side of the femoral bone.

### 3.2. Upper Limb

#### 3.2.1. Craniolateral Approach

When the hindlimb is skinned, the m. quadriceps femoris and the hamstring musculature are obvious (Figure 3A). The m. quadriceps femoris is located on the cranial side of the upper hindlimb and consists of the m. rectus femoris and the mm. vasti, which include the m. vastus medialis, the m. vastus intermedius and the m. vastus lateralis. The m. vastus lateralis can readily be observed using the craniolateral approach. Its pale surface is due to the overlying fascia lata, which is suspended by the m. tensor fasciae latae. This muscle finds its origin on the ilium and the fascia glutea. The latter fascia covers the m. gluteus superficialis and the m. gluteus medius, which are also visible in this view.

Figure 3B shows that the m. vastus lateralis originates from the greater trochanter of the femur. When the m. tensor fasciae latae is detached from the fascia lata and proximally retracted, and the m. quadriceps femoris is separated from the hamstrings, the m. iliopsoas with the n. femoralis on its ventral side can be viewed. The proximal part of the m. rectus femoris, which has its origin just craniodorsal to the acetabulum on the ilium (area m. recti femoris), can be observed cranial to the m. vastus lateralis. On the caudal side, the n. fibularis communis and the n. tibialis, which both run deep to the m. vastus lateralis and the m. biceps femoris in the extension angle of the hip, become visible.

When the m. vastus lateralis is maximally withdrawn from the m. biceps femoris, the m. rectus femoris and the m. vastus intermedius can be examined (Figure 3C). The latter muscle, which was formerly described as the m. crureus, arises from the proximal three-fourths of the shaft of the femur. It is located caudal to the m. rectus femoris and is much vaster. Onto these muscles run muscular branches of the a. glutea caudalis [a. glutea inferior] to supply the hamstring muscles with blood.

The m. rectus femoris and the a., v. and n. femoralis can be better visualized when the m. vastus lateralis is transected in its middle portion and the stumps are proximally and distally retracted (Figure 3D). Like the vasti muscles, the m. rectus femoris inserts into the base of the patella. The femoral artery and vein run toward Scarpa’s triangle on the medial side, while the femoral nerve branches into the m. quadriceps.

#### 3.2.2. Caudolateral Approach

The caudolateral approach to the upper hindlimb allows for examining the hamstring musculature, which is situated caudolateral to the femur, and the sciatic nerve. Removing the skin shows that, from cranial to caudal, the hamstrings consist of the m. biceps femoris, the m. semitendinosus and the m. semimembranosus (Figure 4A). Since the m. biceps femoris and the m. semitendinosus laterally cover the m. semimembranosus, only the proximal portion of the latter muscle is visible.

The transection of the m. biceps femoris and m. semitendinosus permits the study of the m. semimembranosus (Figure 4B). This muscle is composed of the cranial m. semimembranosus accessorius and the caudal m. semimembranosus proprius. The origins and insertions of both were discussed above (Section 3.1.2). The sciatic nerve runs diagonally over the m. semimembranosus accessorius, while muscular branches to the hamstrings are given off.

When the epineurium that surrounds the sciatic nerve is longitudinally incised, the n. fibularis communis and the n. tibialis appear as two separate nerves that arise from the n. ischiadicus close to the extension angle of the hip (Figure 4C). The n. fibularis communis runs parallel to the femur, crosses the m. gastrocnemius caput laterale laterally and finally continues its trajectory in between the m. fibularis longus and the lateral head of the m. gastrocnemius. It gives off branches to the flexors of the tarsus and the extensors of the digits. Caudal to the n. fibularis communis, the n. tibialis courses toward the popliteal cavity, deep in between the lateral and medial heads of the gastrocnemius muscle. Here, it gives off muscular branches to the extensors of the tarsus and the flexors of the digits.

#### 3.2.3. Caudomedial Approach

The medial side of the hindlimb is characterized by the presence of the cranial m. sartorius and the caudal m. gracilis that delineate the trigonum femorale, in which, from cranial to caudal, the n, a. and v. femoralis are situated (Figure 5A). The a. circumflexa femoris lateralis is cranially given off by the initial segment of the a. profunda femoris, where the ln. inguinalis superficialis is positioned. When the m. sartorius is cranially retracted, the m. rectus femoris, the m. vastus medialis, the m. semimembranosus and the m. adductor longus become partly visible. When the femoral artery is followed in the orthograde direction, the a. saphena emerges at the level of the tip of Scarpa’s triangle, while the femoral artery continues its course deep toward the popliteal region. Hence, this artery becomes denominated a. poplitea. Promptly after its emergence, the a. saphena gives off the superficial a. genus descendens [a. genus proximalis], which reaches the medial side of the knee by laterally crossing the sartorius muscle. Analogous to the arterial system, the v. femoralis receives the dual v. saphena medialis [v. saphena magna], which can be followed on either side of the a. saphena. Several anastomoses, crossing over the a. saphena, can be recognized along the trajectory of these venae comitantes cum a. saphena. In the popliteal region, the a. poplitea is accompanied by the v. poplitea, which proximally continues as the v. femoralis at the level of the tip of Scarpa’s triangle, after the v. saphena medialis has been received. The n. femoralis gives off the n. saphenus before ramifying in the m. quadriceps femoris. This superficial nerve joins the a. saphena and the v. saphena medialis.

The hamstring musculature can be examined in detail after the m. sartorius and the m. gracilis are transected and their stumps are retracted (Figure 5B). The m. semimembranosus is composed of the cranial m. semimembranosus accessorius and the caudal m. semimembranosus proprius. As described above, the insertion of the latter is more distal than that of the former. When the hindlimb is examined with a caudomedial approach, the m. semitendinosus is seen more lateral and caudal to the m. semimembranosus. The same is true for the m. biceps femoris in relation to the m. semitendinosus. Retracting the distal stump of the m. gracilis exposes the v. saphena lateralis [v. saphena parva]. This vein drains into the v. caudalis femoris within the popliteal region.

The adductor muscles are studied after the m. semimembranosus and the m. semitendinosus are transected and their stumps are retracted (Figure 5C). The m. adductor longus is located craniomedial to the cranial muscle belly of the m. adductor magnus. The short and thin m. pectineus is medially covered by the m. adductor longus and can be exposed by transecting this muscle. Just distal to the insertion of the adductor longus muscle, the a. femoralis gives off the superficial a. saphena, after which it continues as a. poplitea. When the associated v. poplitea receives the superficial, dual v. saphena medialis that accompanies the a. saphena as venae comitantes, it is denominated v. femoralis. The a. saphena, the v. saphena medialis and the n. saphenus can be followed superficially on the medial side of the hindleg. The popliteal artery and vein run deep in the popliteal region, where the a. poplitea gives off the a. caudalis femoris and the v. poplitea receives the v. caudalis femoris.

Figure 5D demonstrates that the v. saphena lateralis, which runs superficially in the groove formed by the lateral and medial heads of the gastrocnemius muscle, drains into the v. caudalis femoris a few centimeters proximal to the origins of this muscle, i.e., the lateral and medial femoral epicondyles. A sesamoid bone is present in each tendon of origin (ossa sesamoidea m. gastrocnemii or fabellae). The Achilles tendon attaches to the tuber calcanei. The ln. popliteus is located in the proximity of the junction of the v. saphena lateralis and the v. caudalis femoris. The a. circumflexa femoris medialis emerges deep (lateral) to the caudal part of the m. adductor magnus and supplies the m. biceps femoris. By cranially retracting the caudal part of the m. adductor magnus, the m. quadratus femoris becomes exposed with the n. ischiadicus appearing at the distal border of this muscle. A few centimeters more distally, this nerve splits into the cranial n. fibularis communis and the caudal n. tibialis.

### 3.3. Knee

The knee was extensively examined by a caudal approach since several blood vessels and nerves pass through the popliteal region. Lateral, medial and caudomedial will be presented. The knee joint is not included in this topographical description, as its anatomy can be consulted elsewhere [9].

The musculature that is visible directly after skinning the caudal side of the popliteal region consists of the proximally located hamstrings and the gastrocnemius muscle, which forms the superficial muscle mass of the lower leg (Figure 6A). The m. biceps femoris is located laterally, whereas the m. semitendinosus and the m. semimembranosus lie medially. In between the lateral and medial muscles emerge the lateral n. fibularis communis and the medial n. tibialis. In addition, the v. saphena lateralis can be seen draining into the v. caudalis femoris in the vicinity of the ln. popliteus. The lateral saphenous vein, which is located in the groove formed by both heads of the gastrocnemius muscle, has been retracted medially to show the n. cutaneus surae caudalis, a branch of the n. tibialis, in that same groove.

When the semitendinosus, semimembranosus and adductor muscles are removed, the blood vessel ramifications on the caudomedial side of the popliteal cavity can be examined (Figure 6B). The v. saphena lateralis drains into the v. caudalis femoris, which in turn flows into the v. poplitea. A few centimeters more proximal, the v. poplitea receives the superficial v. saphena medialis. This junction gives origin to the v. femoralis. The arterial system follows a similar branching pattern.

The lateral retraction of the m. biceps femoris shows that the n. cutaneus surae lateralis laterally branches off the n. fibularis communis. This cutaneous branch can be biopsied for research purposes. The cutaneous nerve that laterally branches off the n. tibialis is the n. cutaneus surae caudalis. It distally joins the v. saphena lateralis in the groove between both gastrocnemius heads. When these heads are separated, the a. poplitea can be examined in the profundity of the popliteal region. It gives off the a. tibialis cranialis [a. tibialis anterior] and continues as the a. tibialis caudalis [a. tibialis posterior], which joins the n. tibialis. The fine veins that run bilateral to the a. tibialis caudalis and present several anastomoses are the vv. comitantes cum a. tibialis caudalis. Now, the lateral m. soleus and the medial m. flexor digitorum (longus) medialis/tibialis [m. flexor digitorum longus] are also exposed. The m. soleus arises from the head of the fibula. Its tendon fuses with the tendon of the lateral head of the gastrocnemius muscle. The m. flexor digitorum medialis/tibialis, which can be considered the m. flexor digitorum superficialis in domestic mammals, arises halfway from the caudal side of the tibia. Its tendon crosses the medial malleolus and splits to attach to the plantar sides of the distal phalanges of digits II to V.

The deepest layer, which is obtained by the resection of the m. gastrocnemius caput laterale and caput mediale as well as the m. soleus, presents the m. flexor digitorum (longus) lateralis/fibularis [m. flexor hallucis longus] next to the m. flexor digitorum medialis/tibialis. This muscle can be considered the m. flexor digitorum profundus in domestic mammals, as it is located deeper to the latter. It arises from the caudomedial aspect of the fibula, the interosseous membrane between the tibia and fibula, and the distal part of the tibia. Its tendon runs along the plantar side of the tarsal joint and then splits into three tendons, one each for digits I, III and IV. Deep to the m. soleus, lateral to the m. flexor digitorum medialis/tibialis, the m. tibialis caudalis can be observed. This muscle arises from the caudal side of the tibia. Its tendon crosses the medial malleolus and inserts into the plantar sides of the metatarsal bones of digits II to IV.

### 3.4. Lower Limb

#### 3.4.1. Lateral Approach

When the skin of the hindlimb is removed, the following muscles can be recognized from cranial to caudal: m. tibialis cranialis [m. tibialis anterior], m. extensor digiti primi (hallucis) longus, m. extensor digitorum longus, m. fibularis longus and finally the m. gastrocnemius caput laterale (Figure 7A). The m. tibialis cranialis arises from the lateral condyle of the tibia and from the upper two-thirds of its shaft. Two bellies can be observed. The tendon of the medial belly attaches to the first tarsal bone, whereas the tendon of the lateral belly is inserted into the head of the first metatarsal bone. The m. extensor digiti primi (hallucis) longus is proximally covered by the m. extensor digitorum longus but distally gains a more superficial position. This very thin muscle has its origin from the medial side of the fibular diaphysis. The tendon is inserted into the terminal phalanx of the hallux. The m. extensor digitorum longus has two origins, i.e., the lateral condyle of the tibia and the fibular head. Three tendons arise at the level of the foot. The medialmost tendon subsequently divides into two. The resulting four tendons are inserted into the middle and distal phalanges of the 2nd to 5th digits. The m. fibularis longus originates on the fibula and proximal epiphysis of the fibula. The tendon crosses the lateral malleolus and inserts into the plantar side of the first metatarsal bone, thus crossing the plantar side of the foot. The v. marginalis lateralis pedis can be observed on the dorsolateral side of the foot and tarsus. It continues proximally to the tarsus as the v. saphena lateralis, which crosses the tendo calcaneus communis laterally, then runs in the groove formed by the two heads of the gastrocnemius muscle and finally disappears in the popliteal region to drain into the v. caudalis femoris.

The m. biceps femoris and the deep fascia that envelopes the musculature were subsequently removed (Figure 7B). As a result, the individual muscles can now be separated from each other. At the level of the tarsus, two reinforcements of the deep fascia form the retinaculum proximale, which secures the positions of the tarsal flexors and the digital extensors, and the retinaculum distale, which wraps the m. extensor digitorum longus. Proximally, the popliteal artery and vein are visible in the popliteal region. Here, the a. poplitea gives off the a. tibialis cranialis, which runs deep to the m. extensor digitorum longus, and the m. fibularis longus (and brevis) to emerge halfway along the lower limb in the groove formed by the m. tibialis cranialis and the m. extensor digitorum longus. The n. fibularis communis also emerges from the popliteal region, then crosses the lateral head of the m. gastrocnemius and finally splits into the n. fibularis superficialis and the n. fibularis profundus. The n. fibularis superficialis follows the groove between the m. extensor digitorum longus and the m. fibularis longus. The n. fibularis profundus can only be discussed in the deeper layers. The n. tibialis, which is positioned just caudal to the n. fibularis communis in the popliteal region, presents a distal trajectory in between the two heads of the m. gastrocnemius. The a. dorsalis pedis superficialis, which is the superficial branch of the a. saphena, can be seen at the dorsomedial side of the foot.

The deep musculature on the caudal side of the lower limb can be studied after the lateral and medial heads of the gastrocnemius muscle are transected and retracted (Figure 7C). The m. tibialis caudalis [m. tibialis posterior] can now be identified caudal to the m. fibularis longus. This muscle arises from the caudal side of the tibia. Its tendon crosses the medial malleolus and inserts into the plantar sides of the metatarsal bones of digits II to IV. The slender m. plantaris, which has its origin on the lateral femoral condyle and inserts into the plantar fascia by means of a thin tendon that runs along the medial side of the Achilles tendon, is located caudal to the m. tibialis caudalis. Since the proximal half of the m. soleus is laterally covered by the m. plantaris, only its distal segment is visible.

In Figure 7D, the soleus muscle after its transection is shown. This muscle arises from the head of the fibula. Its tendon fuses laterally with the tendon of the gastrocnemius muscle. The retraction of the stumps shows that the proximal half of this muscle is located medial to the m. plantaris, whereas the distal portion is located lateral to the tendons of the plantaris and gastrocnemius muscles.

When the m. tibialis cranialis is pulled cranially and the m. fibularis longus is pulled caudally, the m. extensor digiti primi (hallucis) longus and the m. extensor digitorum longus become fully exposed. In addition, the distal part of the m. fibularis brevis emerges (Figure 7E).

However, the m. fibularis brevis can be described in more detail after the m. extensor digitorum longus is retracted cranially and the m. fibularis longus is pulled caudally (Figure 7F). It arises from the lower two-thirds of the shaft of the fibula and inserts into the metatarsal bone of the fifth digit. The a. tibialis caudalis and the n. fibularis profundus, which both run deep to the m. extensor digitorum longus, can now also be followed in the distal direction.

Like the m. fibularis brevis, the very thin m. fibularis digiti quinti [m. fibularis tertius] arises from the lower two-thirds of the shaft of the fibula. In contrast, it inserts into the metatarsal bone of the fifth digit. Due to its deeper position, it can be seen caudal to the fibularis brevis muscle when this muscle is slightly pulled cranially (Figure 7G).

The m. fibularis longus and the m. plantaris are transected in Figure 7H. As a result, the proximal segment of the tibialis caudalis muscle is visible. Craniolateral to it lie the m. fibularis brevis and the m. fibularis digiti quinti, which indeed arise from the lower two-thirds of the shaft of the fibula.

Finally, the sulcus extensorius of the tibia was exposed after the m. tibialis cranialis was transected and the stumps were retracted (Figure 7I). The proximal stump shows a longitudinal tendinous band that gives origin to the two muscle bellies.

#### 3.4.2. Medial Approach

The muscles that can be identified directly after skinning the medial side of the hindlimb are, from cranial to caudal, the m. tibialis cranialis, the m. flexor digitorum medialis/tibialis and the m. gastrocnemius caput mediale (Figure 8A). The triad consisting of the a. saphena, the v. saphena medialis and the n. saphenus emerges from the distal tip of Scarpa’s triangle, which is formed by the cranial m. sartorius and the caudal m. gracilis. It runs longitudinally on the planum cutaneum tibiae. The a. genus proximalis is given off by the initial segment of the a. saphena. At the level of the tarsus, the a. saphena divides into the superficial a. dorsalis pedis superficialis and the deep a. dorsalis pedis profundus. In this figure, only the former can be seen traveling on the dorsomedial side of the foot.

Removing the deep fascia and separating the individual muscles reveals some muscles that were previously obscured (Figure 8B). The m. popliteus can be seen cranial to the m. gastrocnemius caput mediale and proximal to the m. flexor digitorum medialis/tibialis. The fan-shaped popliteus muscle is the only intrinsic flexor muscle of the knee. It is located on the caudal side of the proximal tibial shaft. Its tendon inserts into the popliteal fossa of the femur. The m. soleus is located deep (lateral) to the m. gastrocnemius caput mediale. The division of the a. saphena into the a. dorsalis pedis superficialis and the a. dorsalis pedis profunda can now be fully distinguished. The v. saphena lateralis can be seen caudal to the hindlimb. It has been caudally pulled out of the groove formed by both heads of the m. gastrocnemius. As regards the ligaments and fascial structures, it is worth mentioning the ligamentum collaterale mediale and the retinaculum proximale.

Transecting the m. gastrocnemius and retracting the stumps reveals the m. plantaris on the caudal side of the m. soleus (Figure 8C). The a. poplitea, which is accompanied by the homonymous vein, continues as the a. tibialis caudalis after the a. tibialis cranialis has branched off. The a. tibialis caudalis is joined by the n. tibialis, deep between the two heads of the m. gastrocnemius.

When the m. soleus is transected and its stumps are retracted, the more caudal (“superficial”) m. plantaris becomes isolated (Figure 8D). This intervention also exposes the m. tibialis caudalis, which is located on the caudolateral side of the m. flexor digitorum medialis/tibialis.

When the m. plantaris is transected as well, and the m. flexor digitorum medialis/tibialis is retracted cranially, the m. flexor digitorum lateralis/fibularis, which is located lateral (“deep”) to the m. flexor digitorum medialis/tibialis, can be visualized (Figure 8E). In addition, the trajectories of the a. tibialis caudalis and the n. tibialis become distinct in between the m. tibialis caudalis on the lateral side and the m. flexor digitorum lateralis/fibularis and the m. flexor digitorum medialis/tibialis on the medial side.

Transecting the m. flexor digitorum medialis/tibialis and retracting its stumps allows for a better visualization of the m. flexor digitorum lateralis/fibularis (Figure 8F). As already mentioned above, the tendon of this muscle runs along the plantar side of the tarsal joint and subsequently splits into three tendons, one each for digits I, III and IV.

### 3.5. Foot

#### 3.5.1. Lateral, Dorsal and Medial Approaches

When the foot is approached laterally, the four tendons of the m. extensor digitorum longus, especially the lateralmost one that inserts into the middle and distal phalanges of the fifth digit, are easily recognizable (Figure 9A). The common tendon of this muscle, thus proximal to its partition, is bridged by the proximal and distal retinacula. The m. fibularis longus, which originates on the tibial head, the interosseous membrane and the proximal epiphysis of the fibula is located lateral to this muscle. After the tendon has traversed the lateral malleolus, it deviates on the plantar side of the foot to insert into the plantar side of the first metatarsal bone. The v. marginalis lateralis pedis drains the dorsal side of the foot and subsequently crosses the tendon of the m. fibularis longus at the level of the lateral malleolus. This vein continues proximally to the tarsus as the v. saphena lateralis. The lateral approach also visualizes some muscles that are located on the plantar side of the foot. These include a portion of the m. flexor digiti quinti brevis, the m. abductor digiti quinti and the m. quadratus plantae.

The four tendons of the m. extensor digitorum longus can easily be discerned from a dorsal view of the foot (Figure 9B). In addition, the two muscle bellies of the m. tibialis cranialis, which each have a separate tendon can be identified. The medial tendon attaches to the first tarsal bone, whereas the lateral tendon is inserted into the base of the first metatarsal bone. The m. extensor digiti primi (hallucis) longus is more obvious compared to the lateral view. After originating from the medial side of the fibular diaphysis, its tendon inserts into the dorsomedial side of the terminal phalanx of the hallux. The m. adductor digiti primi (hallucis) is located in between the hallux and the second digit. This broad muscle has its origin on the plantar sides of the metatarsal bones of the second and third digits. The medial aspect of the proximal phalanx of the hallux is the insertion site. Deep to the m. extensor digitorum longus sits the m. extensor digitorum et digiti primi (hallucis) brevis, which starts at the calcaneus and sends four tendons toward the distal phalanges of digits I to IV. On the lateral side of the foot, the m. abductor digiti quinti, which runs from the tuber calcanei toward the proximal phalanx of the fifth digit, can be recognized. The mm. interossei pedis are arranged in between several metatarsal bones. The arterial system can also be examined in the dorsal view. The a. dorsalis pedis superficialis ramifies superficially, whereas the a. dorsalis pedis profunda continues deep (caudal) to the m. tibialis caudalis to finally reach the plantar side of the foot. The v. marginalis medialis pedis, which drains into the v. saphena medialis, and the v. marginalis lateralis pedis, which drains into the v. saphena lateralis, both drain the dorsal side of the foot.

On the medial view, the paired tendons of the m. tibialis cranialis as well as the tendon of the m. extensor digiti primi (hallucis) longus can be seen at the level of the proximal retinaculum. The medialmost tendons of the m. extensor digitorum longus to the dorsal sides of the second and third digits can also be discerned. On the plantar side, the m. abductor digiti primi (hallucis), which starts from the calcaneus and inserts into the plantar side of the proximal phalanx of the hallux, is partly visible. As also seen from the dorsal side, the m. adductor digiti primi (hallucis) can be seen from the medial view in between the first and second digits. The a. dorsalis pedis superficialis, the a. dorsalis pedis profundus and the v. marginalis medialis pedis can be studied in more detail using the medial approach compared to the dorsal approach.

#### 3.5.2. Plantar Approach

When the plantar side of the foot is studied, the common tendon of the m. flexor digitorum medialis/tibialis can be seen curving around the sustentaculum tali (Figure 10A). This tendon can, however, hardly be followed distally, as it is obscured by several short muscles. Nevertheless, at the level of the metatarsophalangeal joints, the four tendons that originate from that common tendon can be recognized. Thus, the m. flexor digitorum medialis/tibialis is a flexor of digits II to V. Lateral to the common tendon of this flexor muscle sits the m. abductor digiti quinti, which originates on the tuber calcanei and sends a long tendon to the lateral side of the proximal phalanx of the fifth digit. The m. flexor digiti quinti brevis can be observed distal to the abductor muscle of the fifth toe. This muscle has its origin at the tendon of the fibularis longus muscle (remember that the tendon of this muscle crosses the lateral malleolus and inserts into the plantar side of the first metatarsal bone) at the level of the fifth metatarsal bone. It inserts into the proximal phalanx of the fifth digit. On the lateral side of the calcaneus, the m. quadratus plantae originates lateral to the abductor digiti quinti muscle. It splits into several tendons that insert into the tendons of the m. flexor digitorum medialis/tibialis and the m. flexor digitorum lateralis/fibularis. However, these tendons, as well as the tendons of the m. flexor digitorum medialis/tibialis, are obscured by the mm. lumbricales pedis, which represent four fine muscle strands that run medial to the metatarsal bones of the 2nd to 5th digits to insert into their proximal phalanges. The three tendons of the m. flexor digitorum lateralis/fibularis reach the plantar sides of digits I, III and IV. In particular, the tendon to the hallux is substantial. The m. abductor digiti primi (hallucis) is situated on the medial side of the tarsus. As this muscle starts from the medial aspect of the calcaneus, it also obscures the common tendon of the m. flexor digitorum medialis/tibialis. It inserts into the plantar side of the proximal phalanx of the hallux. Distal to the m. abductor digiti primi (hallucis) sits the m. flexor digiti primi (hallucis) brevis. It presents two heads that originate from the plantar side of the tarsus and insert into the proximal phalanx of the hallux. The m. adductor digiti primi (hallucis) is located in between the first and the second digit. It finds its origin on the plantar sides of the second and third metatarsal bones and inserts into the medial side of the proximal phalanx of the hallux. The a. tibialis caudalis presents medial and lateral plantar branches denominated as the a. plantaris medialis and the a. plantaris lateralis. The n. plantaris medialis and n. plantaris lateralis, which are branches of the n. tibialis, can also be recognized on the plantar side of the foot.

The deeper layer clearly shows that the common tendon of the m. flexor digitorum medialis/tibialis splits into four tiny tendons that attach to the plantar sides of the distal phalanges of digits II to V (Figure 10B). The mm. lumbicales sit in between these individual tendons. After transecting these tendons with the associated mm. lumbricales, the mm. contrahentes digitorum pedis and the mm. interossei pedis emerge. The former muscles have a single aponeurosis in common at the level of the fibularis longus tendon. Three muscular bands originate from here to insert into the proximal phalanges of the second, fourth and fifth digits. The deeper interosseus muscles form pairs that are present in each intermetatarsal cleft. They attach to both sides of the metatarsophalangeal joints.

## 4. Discussion

During this anatomical study, remarkable similarities between the rhesus monkey and a human were observed. Since these similarities stretch beyond anatomy and also apply to physiology, the rhesus monkey plays an important role as a model for humans [2,5]. However, the anatomy of the rhesus monkey does not appear to be identical to human anatomy. Several significant discrepancies were observed when comparing the dissected specimens with images in a human anatomy atlas [16]. Access to a state-of-the-art anatomy atlas on the rhesus monkey could therefore be valuable to researchers who seek to extrapolate the results obtained in this nonhuman primate model to humans. Such a publication could also be consulted by veterinarians responsible for the medical care and welfare of rhesus monkeys kept in captivity. These professionals primarily rely on their knowledge of domestic mammal anatomy when treating, for example, wounds or performing surgery on a rhesus monkey. Although the anatomy of this species is remarkably similar to that of domestic mammals, the subtle dissimilarities could result in uncertainty about the optimal care or treatment. Once more, an easily accessible reference work on the rhesus monkey anatomy would be extremely useful.

When the present findings were compared with the literature data on the rhesus monkey anatomy, a few contradictions were noticed. Our dissections revealed that the m. adductor brevis consists of two bellies in the rhesus monkey, whereas the reference works on the anatomy of the rhesus monkey mention the presence of only a single muscle belly [7,8]. These works also state that the v. saphena lateralis drains into the v. poplitea. In contrast, our results indicate that the v. saphena lateralis drains into the v. caudalis femoris, which in turn flows into the v. poplitea. In other words, the v. saphena lateralis does not directly join the v. poplitea. This is also the case in common domestic mammals [13].

When comparing the anatomy of the rhesus monkey with that of common domestic mammals, a number of particularities caught our attention. First, superficial and deep digital flexors, m. flexor digitorum (digitalis in the horse) superficialis and m. flexor digitorum (digitalis in the horse) profundus, respectively, do not exist as such in the rhesus monkey. The m. flexor digitorum (longus) medialis, which has previously been denominated the m. flexor digitorum (longus) tibialis [7]—hence the mention of “m. flexor digitorum (longus) medialis/tibialis” in Section 3—can be regarded as the superficial digital flexor. In domestic mammals, the m. flexor digitorum/digitalis profundus is composed of the m. tibialis caudalis, the m. flexor digitorum/digitalis medialis and the m. flexor digitorum/digitalis lateralis. The latter muscle in the rhesus is the m. flexor digitorum (longus) lateralis, which has previously been termed the m. flexor digitorum (longus) fibularis, hence the mention of “m. flexor digitorum (longus) lateralis/fibularis” in Section 3 [7]. It can be considered the deep digital flexor. The m. tibialis caudalis is a separate muscle in the rhesus monkey and is not one of the muscles that belong to the trigeminal deep digital flexor that is present in domestic mammals. Moreover, it is not a specific digital flexor, but rather a carpal flexor since its tendons insert into the plantar sides of the 2nd to 5th metatarsal bones and not the phalanges. Second, the rhesus monkey presents the m. adductor magnus, longus and brevis. This conformation of the adductor muscles is only seen in the cat. In other domestic mammals, the m. adductor longus has been fused with the m. pectineus [14]. This might be the reason why the m. pectineus is that prominent is those species, while it is rather insignificant in the rhesus monkey and obscured by the m. adductor longus when the hip region is approached medially. Third, the m. semimembranosus consists of two muscle bellies in the rhesus monkey. The smaller and more lateral semimembranosus proprius muscle is inserted medially on the tibial tuberosity, while the larger and more medial semimembranosus accessorius muscle is broadly inserted more proximally, at the level of the medial femoral condyle. This configuration is also seen in the dog [14].

The assessment of similarities and differences between the anatomy of the rhesus monkey and that of a human has been completed at the level of the musculature, the circulatory system, including the arteries and veins, and the nervous system. Table 2 presents the major differences between rhesus monkeys and humans.

First, some peculiarities regarding the muscles are discussed. In the hip region of the rhesus monkey, the m. gluteus medius is more voluminous compared to the m. gluteus superficialis and the m. gluteus profundus. The former is by far the larger out of the three gluteal muscles in humans. Hence, the term m. gluteus maximus is applied [16]. The deep gluteal muscle is denominated m. gluteus minimus in humans and can be absent in this species [8,14,16]. The m. tensor fasciae latae is more voluminous in humans compared to the rhesus monkey [8,16]. The m. psoas minor seems constant in the rhesus monkey, whereas it can be absent in humans [7,8,16]. Both in the rhesus monkey and domestic mammals, the mm. gemelli represent a single muscle, in contrast to what the anatomical term suggests [14]. This term has been adopted from human anatomy, in which the m. gemellus superior and inferior exist [16]. However, one or both mm. gemelli can be absent in humans [7,16].

At the level of the upper limb, both the m. rectus femoris and the m. biceps femoris consist of two heads in humans and only a single head in the rhesus monkey [7,8,16]. Comparable to humans, the equine m. rectus femoris presents a medial and a lateral head. Surprisingly, the equine m. biceps femoris does not contain a single head as in other domestic mammals, nor two heads as in humans—hence the anatomical term biceps—but three heads [14]. The single m. semimembranosus in the horse is partly fused with the m. adductor magnus [14]. In humans, it is the m. semimembranosus accessorius that is fused with m. adductor magnus [7,8,16]. Finally, the human m. pectineus can be composed of two muscle bellies, while only one muscle belly can be discerned in the rhesus monkey [7,16].

The m. tibialis cranialis is located on the cranial side of the lower leg. In contrast to humans, in which the m. tibialis anterior is most often single, the m. tibialis cranialis of the rhesus monkey has two muscle bellies [7,8]. The human m. fibularis longus sends a tendon to the first tarsal bone in addition to the tendon to the first metatarsal bone, which is the sole tendon in the rhesus monkey [7,8]. Both the rhesus monkey and humans possess the m. fibularis brevis, but a third fibularis muscle, i.e., the m. fibularis tertius—hence its name—is present in human beings. Its tendon inserts into the base of the fifth metatarsal bone [16]. It is therefore analogous to the m. fibularis digiti quinti in the rhesus monkey [7]. The presence of the m. fibularis tertius is unique to the horse, which, in contrast to other domestic mammals, lacks the other fibularis muscles [14]. The m. soleus, which, among domestic mammals, can also be identified in the cat and rabbit, consists of two heads in humans and only one in the rhesus monkey, like in the cat and the rabbit [7,8,14,16]. The m. plantaris is more pronounced in the rhesus monkey in comparison with humans [7,8]. The m. flexor digitorum (longus) lateralis/fibularis is, in human anatomy, denoted as the m. flexor hallucis longus since its single tendon inserts into the plantar side of the distal phalanx of the first toe [16], with no additional tendons to the third and fourth toes as seen in the rhesus monkey [7]. Furthermore, the m. flexor digitorum (longus) medialis/tibialis of the rhesus monkey is known as the m. flexor digitorum longus in humans, since this muscle only sends tendons to the 2nd to 5th toes in that species and not to all digits as in the rhesus monkey [7,16].

At the level of the foot, the human m. quadratus plantae comprises two heads instead of one in the rhesus monkey [7,8,16]. The m. flexor digitorum brevis only contains a superficial part in humans. The additional deep part that is seen in the rhesus monkey is absent [7,8,16]. Finally, the mm. contrahentes digitorum pedis are absent in the human foot [16].

The observed differences in the myology of the hindlimb between humans and rhesus monkeys are a reflection of the orthograde posture and associated bipedalism in humans vs. the pronograde posture and associated quadrupedalism in the rhesus monkey [7]. This results in a different position of the center of mass between the two species and in a dissimilar arrangement of the hallux, which, in the rhesus monkey, sits more medially than the anterior human hallux [16].

Besides differences in musculature, the branching pattern of the arterial system of the rhesus monkey slightly deviates from that in humans. In humans, rhesus monkeys and domestic mammals, the a. profunda femoris branches off the proximal segment of the a. femoralis. In humans, the a. profunda femoris gives off the a. circumflexa femoris medialis. This is a typical trait in domestic mammals [13]. Surprisingly, the a. circumflexa femoris medialis of the rhesus monkey is not given off by the a. profunda femoris, but by the a. obturatoria [7]. The a. circumflexa femoris lateralis branches off the a. profunda femoris in both the rhesus monkey and humans [16]. In contrast, the a. circumflexa femoris lateralis is a branch of the a. femoralis in domestic mammals [13].

At the level of the knee, the human femoral artery continues as the popliteal artery, which gives off the a. genus descendens [16]. A similar configuration is seen in domestic mammals [12]. This a. genus descendens is comparable to the a. genus proximalis that is described in the rhesus monkey [7,8]. However, this artery is a branch of the a. saphena, which is given off by the femoral artery. After the a. saphena has branched off, the femoral artery continues as the popliteal artery. Thus, the artery that supplies the craniolateral knee region of the rhesus monkey is not a direct branch of the femoral artery, but a branch of the a. saphena.

The a. saphena is very important in the rhesus monkey, as it plays a pivotal role in the blood supply of the foot, together with the a. tibialis caudalis, which is regarded as the continuation of the popliteal artery in this species [7]. In contrast, the a. saphena is insignificant in humans, in which it is a tiny branch of the a. genus descendens that joins the n. saphenus. It can even be absent in some individuals [7,8,16]. Consequently, the human foot is supplied with blood via the a. tibialis anterior and the a. tibialis posterior [16]. In domestic mammals, the continuation of the popliteal artery is the a. tibialis cranialis. which provides blood to the dorsal side of the foot (a. dorsalis pedis). The plantar side of the foot, but also the dorsal side to some extent, is supplied by the vital a. saphena [13]. Also in humans, the a. dorsalis pedis is in continuation with the a. tibialis anterior [16]. This contrasts with the rhesus monkey, in which the a. dorsalis pedis (both superficialis and profundus) branches off the a. saphena [7,8].

The a. saphena has a venous counterpart called the v. saphena. In fact, the v. saphena is dual. The v. saphena parva in humans drains into the v. poplitea [16]. In mammals, including the rhesus monkey, the v. saphena lateralis first flows into the v. caudalis femoris, which in turn drains into the v. poplitea [13]. The v. caudalis femoris is absent in humans [16]. The v. saphena parva is of clinical importance in the rhesus monkey since it allows for intravenous catheterization or blood collection [9,17].

The other saphenous vein is the v. saphena magna in humans or v. saphena medialis in mammals, including the rhesus monkey. Irrespective of the species, this vein drains into the femoral vein [17,18]. The v. saphena medialis in the rhesus monkey is a paired vein that flanks the a. saphena and, as such, is denominated the vv. comitantes (cum a. saphena). Vv. comitantes were also noticed along the a. tibialis caudalis (vv. comitantes cum a. tibialis caudalis) [17,18]. In humans, vv. comitantes are described alongside deep arteries, but not adjacent to superficial arteries such as the a. saphena [16].

The human terms v. saphena magna and v. saphena parva could be mystifying in the rhesus monkey. Since “magna” means “big, large”, the term v. saphena magna is chosen for that saphenous vein, which is larger in humans. The smaller saphenous vein is termed the v. saphena parva, with “parva” being Latin for “small, tiny”. In the rhesus monkey, however, it is more appropriate to speak of the v. saphena medialis instead of the v. saphena magna and to use the term v. saphena lateralis and not v. saphena parva because, in this species, not the medial but the lateral saphenous vein is larger.

The observed differences between the rhesus monkey and humans regarding the conformation of the nerves are rather trivial. In humans, the n. ischiadicus divides into the n. fibularis communis and the n. tibialis just proximal to the knee [16]. In contrast, the n. ischiadicus is very short in the rhesus monkey. The n. fibularis communis and the n. tibialis can already be recognized at the level of the hip joint after their common epineurium has been incised longitudinally [7,8]. The n. cutaneus surae medialis (known as n. cutaneus surae caudalis in veterinary anatomy) and the n. cutaneus surae lateralis communicate in humans [13,16]. More distally, both nerves join to form the n. suralis [16]. In the rhesus monkey and domestic mammals, no communication is present between the two nerves, and the n. suralis is absent [7,15]. Finally, the m. pectineus is innervated uniquely by the n. femoralis in the rhesus monkey [7], whereas this muscle is additionally innervated by the n. obturatorius in humans [7,16].

## 5. Conclusions

This article provides topographical anatomy data on the hindlimb of the rhesus monkey by means of numerous color images taken at consecutive stages of the dissection. This approach allows for gaining insight into the organization of the anatomical structures in several layers, which might be very helpful during, e.g., surgical interventions. In contrast, previous publications on the anatomy of the rhesus monkey present systematic and not topographic anatomy of the hindlimb, thus describing each system, such as the locomotor and vascular systems, separately or fail to present color images.

The present work facilitates the translation of experimental data obtained in the rhesus monkey to human medicine and might assist veterinarians during, e.g., wound treatment or surgery. In general, the anatomy of rhesus monkeys shows many similarities to the anatomy of humans and domestic mammals. However, we have described the major anatomical differences between the various species. Since these differences are not numerous and are far from substantial, the rhesus monkey appears to be a valuable model for humans.

## Figures and Tables

**Figure 1 vetsci-10-00172-f001:**
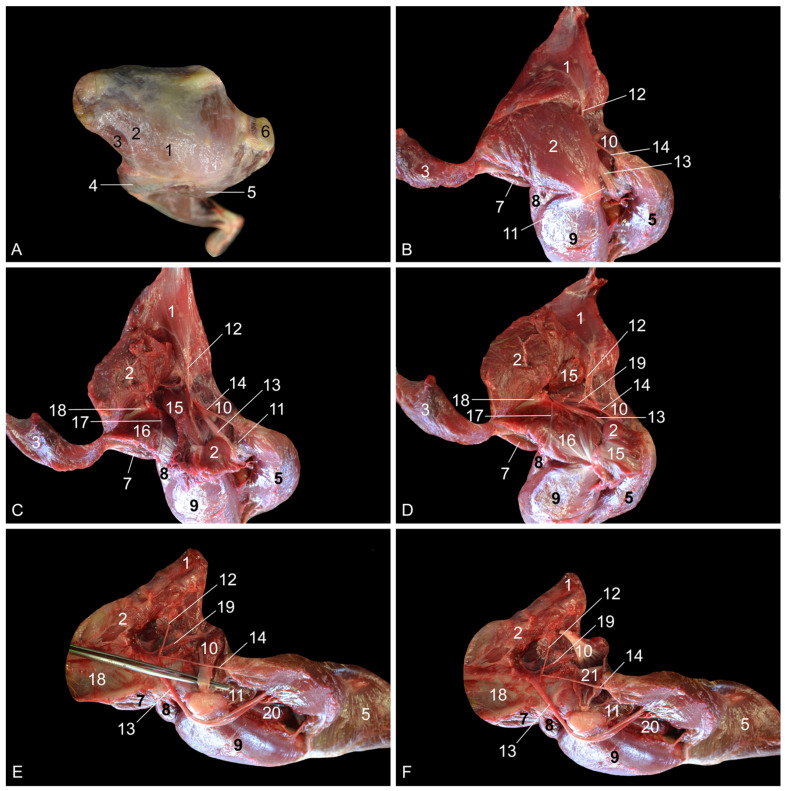
Dorsolateral views of the left hip region. (**A**) Superficial layer; (**B**) second layer after the m. gluteus superficialis and the m. tensor fasciae latae were retracted, and the m. vastus lateralis was detached from the m. biceps femoris; (**C**) third layer after the m. gluteus medius was retracted; (**D**) fourth layer after the m. piriformis was retracted; (**E**) fifth layer after removal of the stump of the m. gluteus profundus, and lateral retraction of the n. ischiadicus; (**F**) deepest layer after the m. obturatorius internus was retracted. 1: M. gluteus superficialis; 2: m. gluteus medius; 3: m. tensor fasciae latae; 4: fascia lata; 5: m. biceps femoris; 6: callositas ischii; 7: m. iliopsoas; 8: m. rectus femoris; 9: m. vastus lateralis; 10: m. obturatorius internus; 11: m. quadratus femoris; 12: n. gluteus caudalis; 13: n. ischiadicus; 14: n. cutaneus femoris caudalis; 15: m. piriformis; 16: m. gluteus profundus; 17: n. gluteus cranialis; 18: corpus ossis ilii; 19: a. pudenda interna + n. pudendus; 20: m. vastus intermedius; 21: mm. gemelli.

**Figure 2 vetsci-10-00172-f002:**
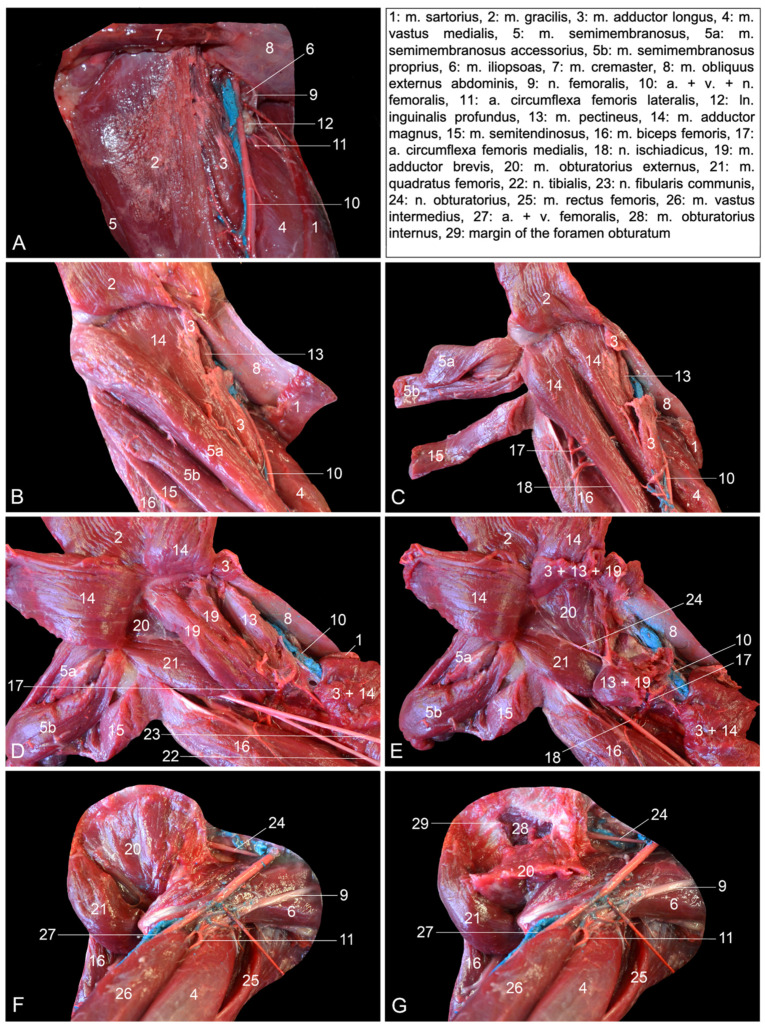
Ventromedial views of the left hip region. (**A**) Superficial layer, in which the m. sartorius is slightly retracted cranially; (**B**) second layer after the m. sartorius and the m. gracilis have been retracted; (**C**) third layer after the m. semimembranosus accessorius, the m. semimembranosus proprius and the m. semitendinosus have been retracted; (**D**) fourth layer after the m. adductor longus and the m. adductor magnus have been retracted; (**E**) fifth layer after the m. pectineus and the m. adductor brevis have been retracted; (**F**) sixth layer after the resection of the muscle stumps of the previously retracted muscles; (**G**) deepest layer after the m. obturatorius externus has been retracted.

**Figure 3 vetsci-10-00172-f003:**
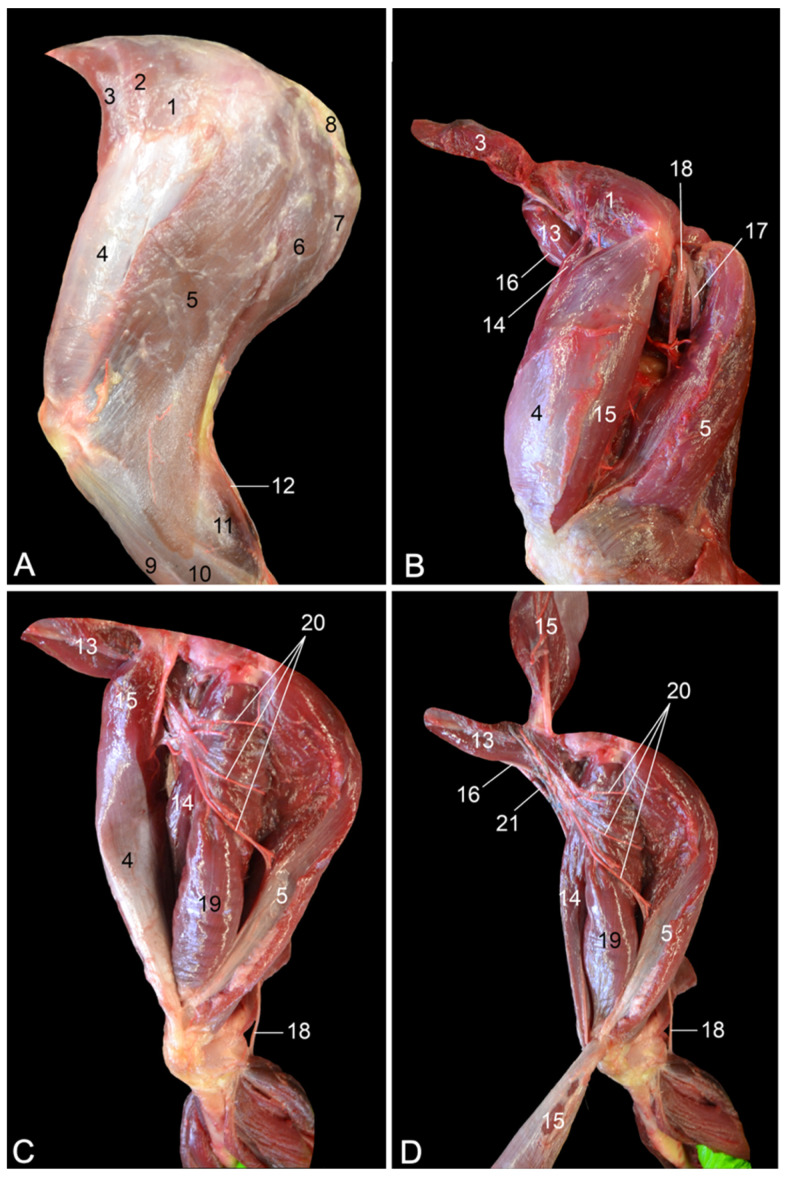
Craniolateral views of the left upper hindlimb. (**A**) Superficial layer; (**B**) second layer after the retraction of the m. tensor fasciae latae and the separation of the m. vastus lateralis from the m. biceps femoris; (**C**) third layer after the m. tensor fasciae latae was removed and the m. vastus lateralis was maximally withdrawn from the m. biceps femoris; (**D**) deepest layer after the m. vastus lateralis was distally retracted. 1: M. gluteus superficialis; 2: m. gluteus medius; 3: m. tensor fasciae latae; 4: fascia lata; 5: m. biceps femoris; 6: m. semitendinosus; 7: m. semimembranosus; 8: callositas ischii; 9: m. tibialis cranialis; 10: m. fibularis longus; 11: m. gastrocnemius caput laterale; 12: v. saphena lateralis; 13: m. iliopsoas; 14: m. rectus femoris; 15: m. vastus lateralis; 16: n. femoralis; 17: n. fibularis communis; 18: n. tibialis; 19: m. vastus intermedius; 20: muscular branches (rami musculares) of the a. glutea caudalis; 21: a. + v. femoralis.

**Figure 4 vetsci-10-00172-f004:**
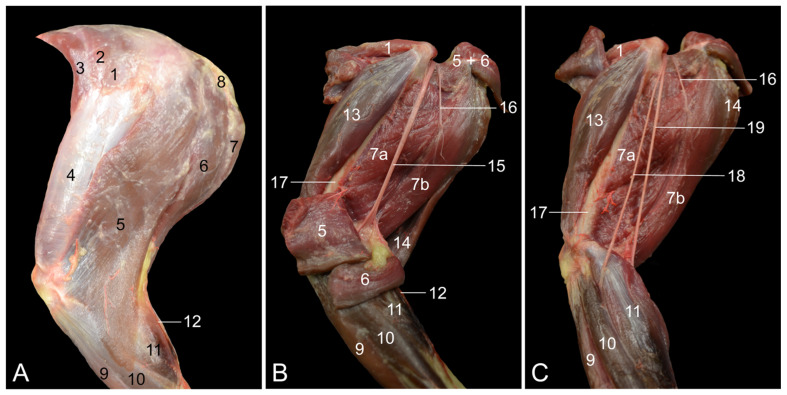
Caudolateral views of the left upper hindlimb. (**A**) Superficial layer; (**B**) second layer after the m. biceps femoris and the m. semitendinosus were transected and their stumps were retracted; (**C**) deepest layer after the stumps of the m. biceps femoris and m. semitendinosus were removed, and the n. fibularis communis was separated from the n. tibialis. 1: M. gluteus superficialis; 2: m. gluteus medius; 3: m. tensor fasciae latae; 4: fascia lata; 5: m. biceps femoris; 6: m. semitendinosus; 7: m. semimembranosus; 7a: m. semimembranosus accessorius; 7b: m. semimembranosus proprius; 8: callositas ischii; 9: m. tibialis cranialis; 10: m. fibularis longus; 11: m. gastrocnemius caput laterale; 12: v. saphena lateralis; 13: m. vastus lateralis; 14: m. gracilis; 15: n. ischiadicus; 16: muscular branches (rami musculares) of the n. ischiadicus; 17: femur; 18: n. fibularis communis; 19: n. tibialis.

**Figure 5 vetsci-10-00172-f005:**
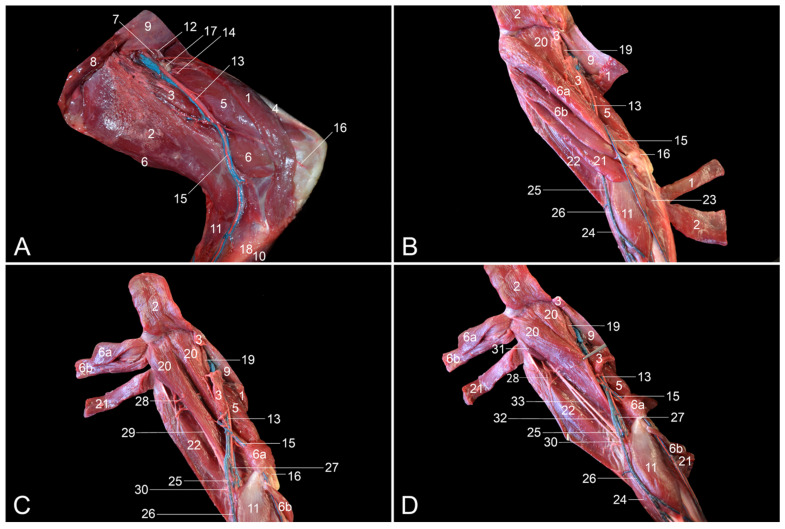
Caudomedial views of the left upper hindlimb. (**A**) Superficial layer in which the m. sartorius is retracted cranially; (**B**) second layer after the m. sartorius and the m. gracilis were transected and their stumps were retracted; (**C**) third layer after the m. semimembranosus accessorius, the m. semimembranosus proprius and the m. semitendinosus were transected and their stumps were retracted; (**D**) deepest layer after the m. adductor magnus and the m. biceps femoris were separated from each other. 1: M. sartorius; 2: m. gracilis; 3: m. adductor longus; 4: m. rectus femoris; 5: m. vastus medialis; 6: m. semimembranosus; 6a: m. semimembranosus accessorius; 6b: m. semimembranosus proprius; 7: m. iliopsoas; 8: m. cremaster; 9: m. obliquus externus abdominis; 10: m. tibialis cranialis; 11: m. gastrocnemius caput mediale; 12: n. femoralis; 13: a. + v. + n. femoralis; 14: a. circumflexa femoris lateralis; 15: a. saphena + v. saphena medialis + n. saphenus; 16: a. genus proximalis; 17: ln. inguinalis superficialis; 18: planum cutaneum tibiae; 19: m. pectineus; 20: m. adductor magnus; 21: m. semitendinosus; 22: m. biceps femoris; 23: m. popliteus; 24: m. gastrocnemius caput laterale; 25: v. caudalis femoris; 26: v. saphena lateralis; 27: a. + v. poplitea; 28: a. circumflexa femoris medialis; 29: n. ischiadicus; 30: ln. popliteus; 31: m. quadratus femoris; 32: n. fibularis communis, 33: n. tibialis.

**Figure 6 vetsci-10-00172-f006:**
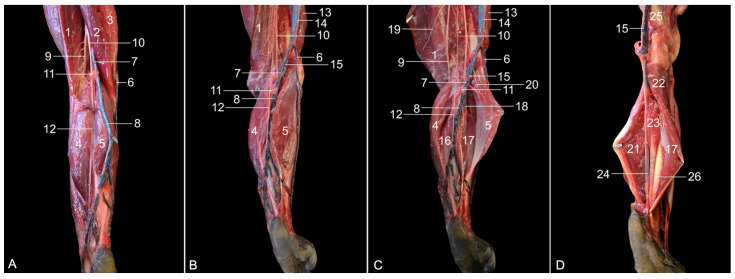
Caudal views of the left popliteal region. (**A**) Superficial layer in which the m. gracilis is removed and the v. saphena lateralis is retracted medially; (**B**) second layer after the m. semitendinosus, the m. semimembranosus and the adductor muscles were resected, showing the natural position of the v. saphena lateralis; (**C**) third layer after lateral retraction of the m. biceps femoris and after the lateral and medial heads of the gastrocnemius muscle were widely splayed; (**D**) deepest layer after the m. gastrocnemius caput laterale and caput mediale and the m. soleus were removed. 1: M. biceps femoris; 2: m. semitendinosus; 3: m. semimembranosus; 4: m. gastrocnemius caput laterale; 5: m. gastrocnemius caput mediale; 6: a. saphena + v. saphena medialis + n. saphenus; 7: v. caudalis femoris; 8: v. saphena lateralis; 9: n. fibularis communis; 10: n. tibialis; 11: ln. popliteus; 12: n. cutaneus surae caudalis; 13: m. vastus intermedius; 14: a. + v. + n. femoralis; 15: a. + v. poplitea; 16: m. soleus; 17: m. flexor digitorum medialis/tibialis; 18: a. tibialis caudalis + vv. comitantes cum a. tibialis caudalis; 19: n. cutaneus surae lateralis; 20: muscular branches (rami musculares) of the n. tibialis; 21: m. tibialis caudalis; 22: m. popliteus; 23: m. flexor digitorum lateralis/fibularis; 24: a. tibialis caudalis + vv. comitantes cum a. tibialis caudalis + n. tibialis; 25: femur; 26: tibia.

**Figure 7 vetsci-10-00172-f007:**
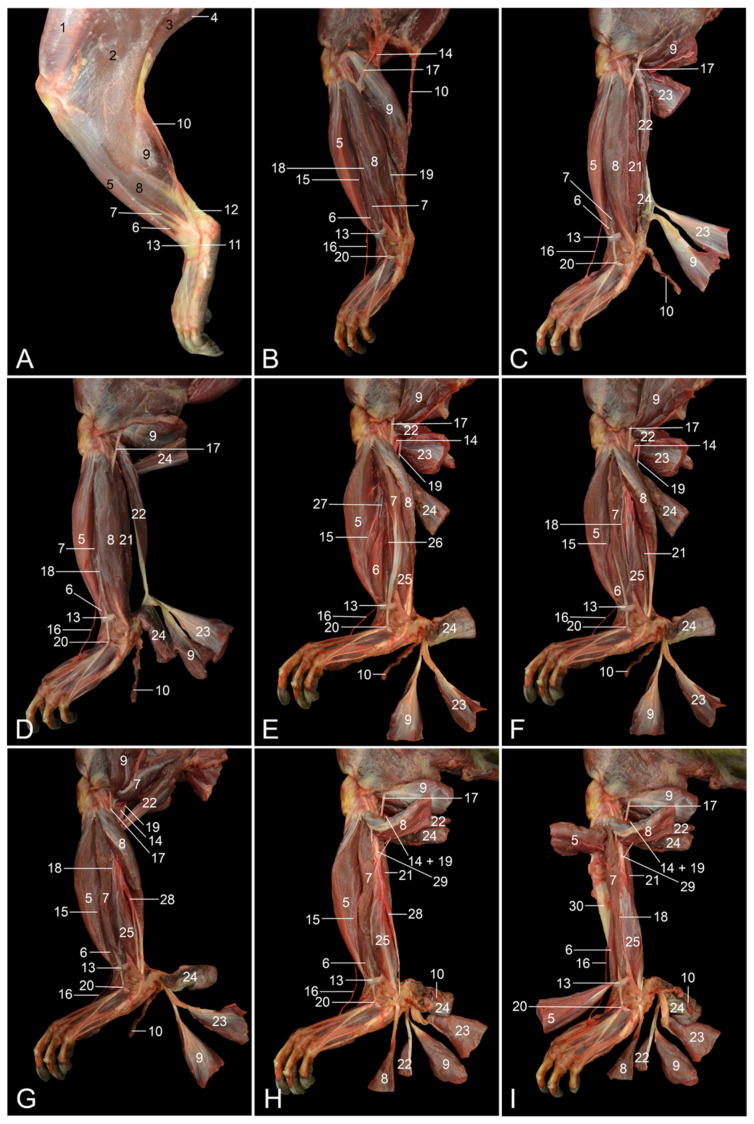
Lateral views of the left lower hindlimb. (**A**) Superficial layer with the superficial venous system filled with red latex; (**B**) second layer after the m. biceps femoris was retracted; (**C**) third layer after the m. gastrocnemius caput laterale, the m. gastrocnemius caput mediale and the v. saphena lateralis were transected and retracted; (**D**) fourth layer after the m. soleus was transected and the stumps were retracted; (**E**) fifth layer after the m. plantaris was transected and the stumps were retracted; (**F**) sixth layer after cranial retraction of the m. extensor digitorum longus and the m. fibularis brevis; (**G**) seventh layer after the m. fibularis longus and the m. fibularis brevis were separated; (**H**) eighth layer after the m. fibularis longus was transected and the stumps were retracted; (**I**) deepest layer after the m. tibialis cranialis was transected and the stumps were retracted. 1: Fascia lata; 2: m. biceps femoris; 3: m. semitendinosus; 4: m. semimembranosus; 5: m. tibialis cranialis; 6: m. extensor digiti primi (hallucis) longus; 7: m. extensor digitorum longus; 8: m. fibularis longus; 9: m. gastrocnemius caput laterale; 10: v. saphena lateralis; 11: v. marginalis lateralis pedis; 12: tendo calcaneus communis; 13: retinaculum proximale; 14: a. + v. poplitea; 15: a. tibialis cranialis; 16: a. dorsalis pedis superficialis; 17: n. fibularis communis; 18: n. fibularis superficialis; 19: n. tibialis; 20: retinaculum distale; 21: m. tibialis caudalis; 22: m. plantaris; 23: m. gastrocnemius caput mediale; 24: m. soleus; 25: m. fibularis brevis; 26: a. tibialis caudalis; 27: n. fibularis profundus; 28: m. fibularis digiti quinti; 29: fibula; 30: tibia.

**Figure 8 vetsci-10-00172-f008:**
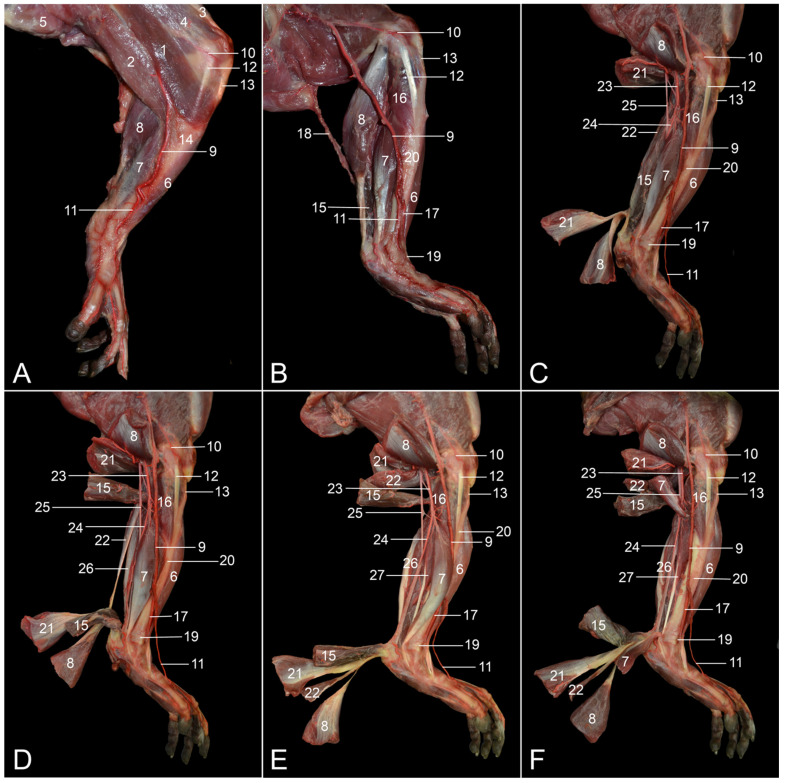
Medial views of the left lower hindlimb. (**A**) Superficial layer; (**B**) second layer after the m. sartorius and the m. gracilis were removed and the m. flexor digitorum medialis/tibialis was separated from the m. gastrocnemius caput mediale; (**C**) third layer after the m. gastrocnemius caput laterale, the m. gastrocnemius caput mediale and the v. saphena lateralis were transected and retracted; (**D**) fourth layer after the m. soleus was transected and its stumps were retracted; (**E**) fifth layer after the m. plantaris was transected and its stumps were retracted; (**F**) deepest layer after the m. flexor digitorum medialis/tibialis was transected and its stumps were retracted. 1: M. sartorius; 2: m. gracilis; 3: m. rectus femoris; 4: m. vastus medialis; 5: m. semimembranosus; 6: m. tibialis cranialis; 7: m. flexor digitorum medialis/tibialis; 8: m. gastrocnemius caput mediale; 9: a. saphena + v. saphena medialis + n. saphenus; 10: a. genus proximalis; 11: a. dorsalis pedis superficialis; 12: ligamentum collaterale mediale; 13: ligamentum patellae; 14: planum cutaneum tibiae; 15: m. soleus; 16: m. popliteus; 17: a. dorsalis pedis profunda; 18: v. saphena lateralis; 19: retinaculum proximale; 20: tibia; 21: m. gastrocnemius caput laterale; 22: m. plantaris; 23: a. + v. poplitea; 24: a. tibialis caudalis; 25: n. tibialis; 26: m. tibialis caudalis; 27: m. flexor digitorum lateralis/fibularis.

**Figure 9 vetsci-10-00172-f009:**
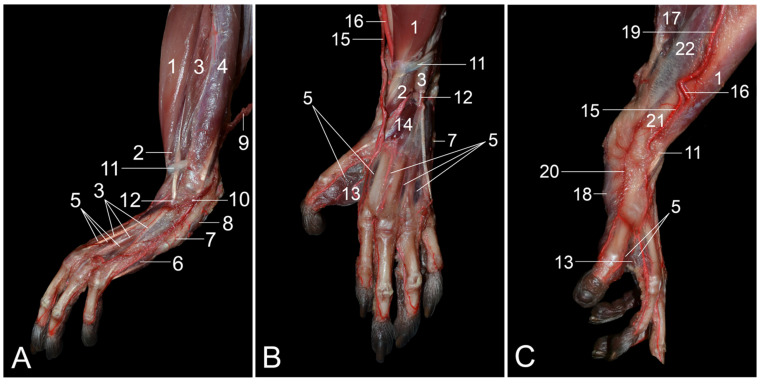
Lateral (**A**), dorsal (**B**) and medial (**C**) views of the left foot. 1: M. tibialis cranialis; 2: m. extensor digiti primi (hallucis) longus; 3: m. extensor digitorum longus; 4: m. fibularis longus; 5: mm. interossei pedis; 6: m. flexor digiti quinti brevis; 7: m. abductor digiti quinti; 8: m. quadratus plantae; 9: v. saphena lateralis; 10: v. marginalis lateralis pedis; 11: retinaculum proximale; 12: retinaculum distale; 13: m. adductor digiti primi (hallucis); 14: m. extensor digitorum et digiti primi (hallucis) brevis; 15: a. dorsalis pedis superficialis; 16: a. dorsalis pedis profunda; 17: m. gastrocnemius caput mediale; 18: m. abductor digiti primi (hallucis); 19: a. saphena + v. saphena medialis + n. saphenus; 20: v. marginalis medialis pedis; 21: planum cutaneum tibiae; 22: m. flexor digitorum medialis/tibialis.

**Figure 10 vetsci-10-00172-f010:**
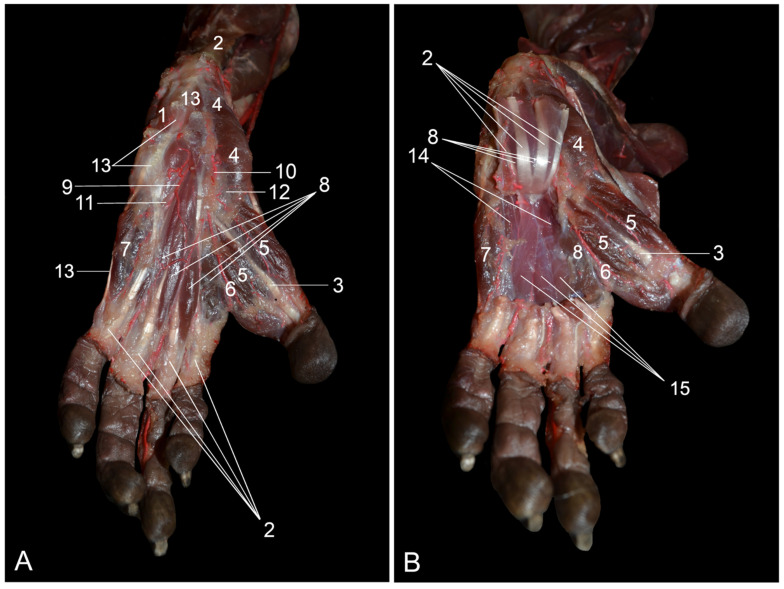
Plantar views of the left foot. (**A**) Superficial layer; (**B**) deeper layer after the tendons of the m. flexor digitorum medialis/tibialis and the mm. lumbricales pedis were proximally retracted. 1: m. quadratus plantae; 2: m. flexor digitorum medialis/tibialis; 3: m. flexor digitorum lateralis/fibularis; 4: m. abductor digiti primi (hallucis); 5: m. flexor digiti primi (hallucis) brevis; 6: m. adductor digiti primi (hallucis); 7: m. flexor digiti quinti brevis; 8: mm. lumbricales pedis; 9: a. plantaris lateralis; 10: a. plantaris medialis; 11: n. plantaris lateralis; 12: n. plantaris medialis; 13: m. abductor digiti quinti; 14: mm. interossei pedis; 15: mm. contrahentes digitorum pedis.

**Table 1 vetsci-10-00172-t001:** Overview of the presented regions of the pelvic limb of the rhesus monkey.

3.1.	Hip region			
3.1.1.		lateral approach		
3.1.2.		medial approach		
3.2.	Upper Limb			
3.2.1.		craniolateral approach		
3.2.2.		caudolateral approach		
3.2.3.		caudomedial approach		
3.3.	Knee			
3.4.	Lower limb			
3.4.1.		lateral approach		
3.4.2.		medial approach		
3.5.	Foot			
3.5.1.		lateral, dorsal and medial approach		
3.5.2.		plantar approach		

**Table 2 vetsci-10-00172-t002:** Differences observed between the anatomy of the hindlimb of the rhesus monkey and the anatomy of the human leg.

Rhesus Monkey	Human
m. gluteus superficialis	m. gluteus maximus
m. gluteus profundus	m. gluteus minimus
m. tensor fasciae latae: weak	m. tensor fasciae latae: heavy
mm. gemelli: single	m. gemellus superior et inferior
m. rectus femoris: single	m. rectus femoris: double
m. biceps femoris: single	m. biceps femoris: double
m. pectineus: single	m. pectineus: often double
m. tibialis cranialis: double	m. tibialis anterior: single
m. fibularis longus: tendon to os metatarsale I	m. fibularis longus: additional tendon to os tarsale I
m. fibularis digiti quinti	m. fibularis tertius
m. soleus: single	m. soleus: double
m. plantaris: heavy	m. plantaris: weak
m. flexor digitorum (longus) lateralis/fibularis: 3 tendons	m. flexor hallucis longus: single tendon
m. flexor digitorum (longus) medialis/tibialis: 5 tendons	m. flexor digitorum longus: 4 tendons
m. quadratus plantae: single	m. quadratus plantae: double
m. flexor digitorum brevis: superficial and deep part	m. flexor digitorum brevis: only superficial part
mm. contrahentes digitorum pedis: present	mm. contrahentes digitorum pedis: absent
a. circumflexa femoris medialis: branch of a. obturatoria	a. circumflexa femoris medialis: branch of a. profunda femoris
a. genus proximalis: branch of a. saphena	a. genus descendens: branch of a. poplitea
a. saphena: prominent, branch of a. femoralis	a. saphena: insignificant, branch of a. genus descendens
a. dorsalis pedis: branch of a. saphena	a. dorsalis pedis: continuation of a. tibialis anterior
v. saphena lateralis: large, drains into v. caudalis femoris	v. saphena parva: small, drains into v. poplitea
v. saphena medialis: small, paired (vv. comitantes cum a. saphena)	v. saphena magna: large, single
n. cutaneus surae medialis et lateralis are independent	n. cutaneus surae medialis et lateralis form n. suralis

## Data Availability

Not applicable.

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
