# Peer review of "Topographical Anatomy of the Rhesus Monkey (Macaca mulatta)—Part II: Pelvic Limb"

_vetsci, 2023, doi:10.3390/vetsci10030172_

Round 1

Reviewer 1 Report

General comments

The manuscript describes in detail aspects about topographical anatomy of the rhesus monkey (Macaca mulatta) pelvic limb

 The topic of this article is within the scope of the journal and represents a major field of research. It is a descriptive study; however, I suggest describing the ages of the animals, as well as highlighting the ethical origin of the individuals for the development of the study, taking into account that some of the authors have already published a text with anatomical descriptions of this same species.

Anatomy of the Rhesus Monkey (Macaca mulatta): The Essentials for the Biomedical Researcher

WRITTEN BY Christophe Casteleyn and Jaco Bakker

Reviewed: June 25th, 2021 Published: September 13th, 2021

DOI: 10.5772/intechopen.99067

Below my indications:

ABSTRACT

It is well written. It briefly describes the objective, relevant results and conclusions. the study is justified

INTRODUCTION

It describes relevant aspects of the subject to be analyzed that justify carrying out the work. However, I suggest including another objective "to identify the similarities and differences in the region between the human anatomy and that of the Rhesus monkey" as it is developed in the discussion section.

MATERIALS AND METHODS

Although in an annex the authors add the legend about not requiring documents from the ethics committees of their institutions, it is suggested highlighting the ethical origin of the individuals for the development of the study, taking into account that some of the authors have already published a text with anatomical descriptions of this same species.

Line 79

2.1 Animals. I recommend adding the age of the animals, just to know if there is any difference in terms of sizes or volumes of the structures studied.

RESULTS

In this section, the text describes the anatomical regions and they are accompanied by allusive images with their explanation in the image caption.

DISCUSSION

It is well written. It describes similarities and differences with human anatomy. I suggest including a comparative table with the main findings in a concise way and for greater detail, the text could be reviewed where they are described in greater depth.

CONCLUSIONS

It is well written. It describes similarities and differences with human anatomy, however, I suggest that the authors highlight the impact of the anatomical description of the region and the contributions of their work to the anatomy of Rhesus monkeys compared to other studies.

REFERENCES

The references used are  related to the study to be developed.

Reviewer 2 Report

The article is very interesting. Undoubtedly, this material will be very interesting and important for anatomists. The description of the anatomy of the rhesus monkey is written here more like about for normal anatomy. In topographic anatomy, we the anatomists use transverse sections of limbs. They just show the entire topographic anatomy of the regions. The photographs here are of high quality but show normal anatomy.  When we study topography, we show all foramina and canals. It is not shown foramen suprapiriformes and for. infrapiriformes (fig 1 – c, d). If they are in Macaca mulatta.  It would be very good if the authors add to the tables the differences in the anatomy of the Macaca mulatta and humans. The differences in anatomy shown in the tables will be very convenient to use for us. It would also be very good if the authors make drawings (schemes). Transverse-section drawings or schemes (topographic anatomy) and for normal anatomy too.

Reviewer 3 Report

The Authors describe the anatomy of the pelvic climb in the Rhesus monkey, in order to give an useful reference for clinical and surgical practices performed in this animal model. The paper is of very high quality, especially for the numerous detailed full-color pictures.

Following some suggestions (with a question), mostly graphic, considering that there is not points in need of revision.

Line 103: a question about the technique of latex injection which may possibly require an ethical approval, depending on the method adopted; according to the sequence of methodolgical approach as described in the text,  it seems that the injection was performed after defrosting the cadavers; if it’s so, how the Authors succeeded (e.g. avoiding cloth formation or other dfrawbacks) considering that the cited reference 11 (Cornillie at al.) suggest that the specimen to be casted should be as fresh as possible, eventually heparinizing the animal before euthanasia?

Line 121: it lacks in the list of References (see page 28) the references 20 and 21.

Line 130: correct in “regiones genus and poplitea” or “regio genus and regio poplitea”

Lines 260 and everywhere in the text (lines 411, 417, 457): consider to put “a few centimetres” instead “a few cm”.

Line 270: correct in “In Figures 2 F and G”

Line 363: for not mixing latin and english (avoiding to put all in Latin), correct in “muscular branches (rami musculares) of n. ischiadicus”

Lines 412 and 549: correct in “Achilles tendon” or in latin “Achillis tendinis”

Line 636: “the m. flexor….” without “de”

Line 726 “(remember that….)”

Line 827: between capitals put the correct reference of human atlas, perhaps 16.

Line 923: between capital [16] without “m”.
